# The Phosphatidylserine Receptor TIM-1 Enhances Authentic Chikungunya Virus Cell Entry

**DOI:** 10.3390/cells10071828

**Published:** 2021-07-20

**Authors:** Jared Kirui, Yara Abidine, Annasara Lenman, Koushikul Islam, Yong-Dae Gwon, Lisa Lasswitz, Magnus Evander, Marta Bally, Gisa Gerold

**Affiliations:** 1Centre for Experimental and Clinical Infection Research, TWINCORE, Institute for Experimental Virology, a Joint Venture between the Medical School Hannover and the Helmholtz Centre for Infection Research, 30625 Hannover, Germany; Jared.Kirui@tiho-hannover.de (J.K.); anna-sara.lenman@umu.se (A.L.); Lisa.Lasswitz@tiho-hannover.de (L.L.); 2Department of Biochemistry & Research Center for Emerging Infections and Zoonoses (RIZ), University of Veterinary Medicine Hannover, 30559 Hannover, Germany; 3Department of Clinical Microbiology, Umeå University, 90185 Umeå, Sweden; yara.abidine@umu.se (Y.A.); islam.koushikul@umu.se (K.I.); kwon.yongdae@umu.se (Y.-D.G.); magnus.evander@umu.se (M.E.); marta.bally@umu.se (M.B.); 4Wallenberg Centre for Molecular Medicine (WCMM), Umeå University, 90185 Umeå, Sweden

**Keywords:** Chikungunya virus, CHIKV, alphavirus, enveloped virus, phosphatidylserine, T-cell immunoglobulin and mucin domain 1, TIM-1, Axl receptor tyrosine kinase, Axl, entry

## Abstract

Chikungunya virus (CHIKV) is a re-emerging, mosquito-transmitted, enveloped positive stranded RNA virus. Chikungunya fever is characterized by acute and chronic debilitating arthritis. Although multiple host factors have been shown to enhance CHIKV infection, the molecular mechanisms of cell entry and entry factors remain poorly understood. The phosphatidylserine-dependent receptors, T-cell immunoglobulin and mucin domain 1 (TIM-1) and Axl receptor tyrosine kinase (Axl), are transmembrane proteins that can serve as entry factors for enveloped viruses. Previous studies used pseudoviruses to delineate the role of TIM-1 and Axl in CHIKV entry. Conversely, here, we use the authentic CHIKV and cells ectopically expressing TIM-1 or Axl and demonstrate a role for TIM-1 in CHIKV infection. To further characterize TIM-1-dependent CHIKV infection, we generated cells expressing domain mutants of TIM-1. We show that point mutations in the phosphatidylserine binding site of TIM-1 lead to reduced cell binding, entry, and infection of CHIKV. Ectopic expression of TIM-1 renders immortalized keratinocytes permissive to CHIKV, whereas silencing of endogenously expressed TIM-1 in human hepatoma cells reduces CHIKV infection. Altogether, our findings indicate that, unlike Axl, TIM-1 readily promotes the productive entry of authentic CHIKV into target cells.

## 1. Introduction

Chikungunya fever, caused by chikungunya virus (CHIKV), has emerged as a global health problem in the last seven decades [1,2]. CHIKV is an arbovirus and member of the Togaviridae family, genus *Alphavirus* transmitted to humans mainly by *Aedes (Ae.) aegypti* and *Ae. albopictus* mosquitoes [3]. The species CHIKV consists of three main genotypes, namely East-Central-South-African (ECSA), West African, and Asian [4]. It is estimated that about 75–95% of infected individuals develop chikungunya fever, with symptoms such as high fever, intense asthenia, myalgia, rash, and debilitating joint pain that turns chronic in 12–49% of patients [5,6]. Therapeutic options for CHIKV are limited since there are currently no specific antivirals and no licensed vaccines.

CHIKV has a wide cellular and tissue tropism which may be attributed to use of ubiquitously expressed molecules or several cell specific factors for entry. These molecules likely determine CHIKV pathogenesis and represent promising targets for antiviral strategies [7,8,9,10,11]. Multiple attachment factors and putative receptors for CHIKV and other alphaviruses have been documented [12,13]. For instance, ATP synthase β subunit (ATPSβ) is a host factor in mosquito cells [14] and prohibitins [15], glycosaminoglycans [16,17], phosphatidylserine (PtdSer)-mediated virus entry-enhancing receptors (PVEERs) [18,19], and MXRA8 [20] are host factors in mammalian cells. Interaction with the cell surface molecules is mediated by the viral E2 glycoprotein, whose domain B contains receptor binding sites [17,21]. For MXRA8, a recently identified receptor for several alphaviruses [20] E1 is additionally important as MXRA8 engages amino acid residues at the E1 and E2 glycoprotein heterodimer interface [22,23]. Phagocytic cells express PVEERs through which they bind PtdSer present on the outer leaflets of apoptotic bodies [24,25,26]. Similarly, epithelial cells expressing T-cell immunoglobulin and mucin domain 1 (TIM-1) act as semi-professional phagocytes and are involved in the clearance of apoptotic bodies [27,28], a process mediated by phosphorylation of residues in the cytoplasmic domain of TIM-1 [29]. Some enveloped viruses have evolved to incorporate PtdSer in the viral membrane, hence disguised as apoptotic bodies, a phenomenon termed as apoptotic mimicry [19,30,31]. TIM-1 and Axl receptor tyrosine kinase (Axl) are PVEERs associated with enhanced cell entry by enveloped viruses. This includes alphaviruses, filoviruses, and flaviviruses, among others [18,19,32,33,34]. Using CHIKV glycoprotein based pseudoviruses, the TIM family of proteins and Axl were shown to enhance infection [19,33]. TIM-1 and Axl are single pass transmembrane proteins with distinct ectodomains and cytoplasmic domains. TIM-1 interacts with PtdSer through a binding pocket known as metal ion ligand binding site (MILIBS) on its extracellular immunoglobulin-like variable (Ig-V) domain [33]. Axl indirectly binds phosphatidylserine through ligands, namely growth arrest-specific factor 6 (Gas6) [35] or protein S1 (ProS1) [36]. In the skin, TIM-1 and Axl are predominantly expressed by keratinocytes in the basal layer of the epidermis [37,38]. HaCat cells derived from spontaneously immortalized keratinocytes serve as a relevant model to study keratinocytes in vitro [39]. However, they hardly express TIM-1 and Axl. The role of TIM-1 and Axl expression in permissiveness of keratinocytes is yet to be characterized.

After binding to a receptor on the plasma membrane, CHIKV primarily enters human host cells by clathrin-mediated endocytosis [40,41]. However, clathrin-independent pathways have also been reported [8,42]. Upon endocytosis, CHIKV particles are delivered to early endosomes in mammalian cells [42], whereas in mosquito cells, the complexes traffic further to maturing or late endosomes before membrane fusion occurs [43]. The discrepancy in the endosomal fusion compartments may be due to the variability of endosomal cues between cells. Nonetheless, general molecular mechanisms involved in fusion are highly conserved between alphaviruses. Specifically, the acidic endosomal environment triggers a class II membrane fusion mechanism [44,45] and release of the nucleocapsid into the cytosol [46,47].

In the current study, we have examined the role of TIM-1 and Axl in CHIKV infection using different genotypes of authentic virus. Our experiments show that TIM-1 unlike Axl is functional as an entry factor for CHIKV and that the PtdSer binding site as well as the cytoplasmic domain are essential for infection. These results indicate that CHIKV exploits the apoptotic cell clearance pathway to facilitate the rapid and efficient infection of human cells.

## 2. Materials and Methods

### 2.1. Cells and Viruses

Human embryonic kidney 293T (HEK293T) cells [48] obtained from American Type Culture Collection (ATCC, CRL-3612), baby hamster kidney cells (BHK-21, ATCC CCL-10), human hepatoma derived (Huh7.5) cells [49] (kindly provided by Charles Rice, Rockefeller University, New York, NY, USA), spontaneously immortalized human skin keratinocytes (HaCat cells) and dermal fibroblasts were kindly provided by PD Dr. F. Pessler, Twincore, Hannover and Vero cells (ATCC CRL-1586) were cultured in Dulbecco’s modified essential medium (DMEM, Gibco™, Paisley, Scotland, UK) supplemented with 10% fetal calf serum (FCS, Gibco™), 100 U/mL penicillin, 100 µg/ml streptomycin, 1% non-essential amino acids and 2 mM L-glutamine, at 37 °C in 5% CO_2_ humidified incubator_._ Chinese hamster ovary (CHOK1 and CHO745) cells obtained from ATCC were cultured in RPMI-1640 (Gibco™, Paisley, Scotland, UK) supplemented with 10% fetal calf serum (FCS, Gibco™), 100 U/mL penicillin, 100 µg/ml streptomycin, 1% non-essential amino acids, and 2 mM L-Glutamine. 

The chikungunya virus strains; East Central South African (ECSA) LR2006-OPY1 strain (3′GFP-CHIKV) [50] and West African (WA) 37997 strain (5′GFP-CHIKV) [51], both encoding green fluorescent protein (GFP) gene, and Asian (181/25) vaccine strain [52] encoding either mCherry-fluorescent protein (mc-CHIKV) or nano-luciferase gene (nLuc-CHIKV) fused to the N-terminus of E2 glycoprotein (Appendix A). The mCherry or nano-luciferase proteins are expressed on the viral envelope as previously described [53]. The plasmids used for the production of CHIKV were kindly provided by Graham Simmons, San Francisco, CA, USA. The live attenuated Venezuelan equine encephalitis virus (VEEV) TC-83 strain expressing a GFP reporter gene [54] was generated by Mike Diamond, Saint Louis, MO, USA. Sindbis virus, Lövånger (KF737350.1) strain [55] was provided by Magnus Evander, Umeå, Sweden. The GFP expressing human adenovirus type HAdV-C5 (HAdV-5) was obtained from Vector Development Laboratory, Houston, TX, USA.

### 2.2. Plasmids and Antibodies

Gene fragments (gBlocks) encoding wild type TIM-1/Axl and respective mutant open reading frames were commercially synthesized by Integrated DNA Technologies (IDT, Inc., Coralville, IA, USA). The gene fragments were amplified by PCR and cloned into pWPI_BLR vector using the Gibson assembly method according to the manufacturer’s instructions (New England Biolabs, Ipswich, MA, USA) and direct sequencing used to confirm inserts.

TIM-1 polyclonal (TIM-1 pAb, AF1750) and Axl polyclonal (Axl pAb) antibodies were purchased from R&D systems while TIM-1 monoclonal antibody (TIM-1 mAb) was purchased from BioLegend^®^, San Diego, CA, USA.

### 2.3. RNA Transfection by Electroporation

HEK293T cells (1 × 10^6^) were resuspended in 400 µL of cytomix electroporation buffer (2 mM ATP, 5 mM glutathione, 120 mM KCl, 0.15 mM CaCl_2_, 10 mM K_2_HPO_4_/KH_2_PO_4_ (pH 7.6), 25 mM HEPES, 2 mM of EGTA and 5 mM MgCl_2_). Either CHIKV sub-genomic replicon (SGR) or full-length CHIKV genome encoding nanoluciferase gene were added at a concentration of 1 µg and 1.7 µg respectively. The mixture was transferred to a 0.4 cm sterile cuvette and electroporated at 240 V and 975 Ω using a Bio-Rad electroporator system. After electroporation, cells were gently resuspended in 2 mL of prewarmed DMEM supplemented with 10% FCS, 2 mM L-Glutamine. Cells were seeded in duplicates in 96 and 24-well plates for SGR and the full-length RNA assays respectively. The cells were cultured at 37 °C and 5% CO_2_. At the indicated timepoints, the cells were lysed for luciferase assay.

### 2.4. Generation of Lentiviral Vectors and Transduction of Cells 

To generate lentiviral pseudoparticles, HEK293T cells were co-transfected with three plasmids. For pseudoparticles used to generate cells stably expressing a protein of interest, the cells were transfected with pVSV-G encoding the G protein for the Vesicular stomatitis virus (VSV), the lentiviral packaging plasmid pCMV_ΔR8-74 and the pWPI (from Didier Trono (Addgene plasmid # 12254) encoding either the wild type or mutant (TIM-1 or Axl) and a blasticidin resistance gene. Sodium butyrate was added 24 h post transfection in order to boost plasmid transcription [56]. For cell entry experiments, the pseudoparticles were generated using a plasmid encoding for a glycoprotein of the virus of interest (CHIKV, EBOV or VSV), the lentiviral packaging plasmid pCMV_ΔR8-74, and a pWPI plasmid encoding a luciferase gene as a reporter protein. At 48 and 72 h post transfection, lentiviral particles were harvested by filtering the supernatant through a 0.45 μm pore size filter. The lentiviral particles were stabilized by adding HEPES and polybrene was added to improve the efficiency of gene transfer [57]. 

To generate cells stably expressing TIM-1 or Axl, the particles were added to a monolayer of cells for five hours of transduction then replenished with fresh media. Selection for positively transduced cells with blasticidin (5 µg/mL) commenced 48 h post-transduction. To determine the role of surface proteins in virus cell entry experiments, cells were transduced with lentiviral pseudoparticles for 4 h and incubated with fresh media for 24 h.

### 2.5. RNA Interference 

Huh7.5 cells pre-seeded in 6-well plates for five hours were transiently transfected with a pool of three siRNAs (Ambion™ Silencer™ Select) for TIM-1 (s230290, s230291, s25632) and MXRA8 (s29242, s29241, s29240) and a control non-targeting (NT) siRNA (AM4637) (ThermoFisher) using Lipofectamine RNAiMAX Reagent protocol (ThermoFisher, Waltham, MA, USA). At 48 h post-transfection, cells were assessed for expression, then seeded for infection and viability testing. The cells were infected with CHIKV at the indicated MOI and susceptibility determined by flow cytometry at 24 h post infection. 

### 2.6. Cell Viability and Proliferation Assay

The cellular metabolic activity was measured using the MTT assay as previously described [58]. Cells were seeded in 96-well plates at a density of 2 × 10^4^ cells/well. Medium was replaced with 50 µL of 0.5 mg/mL MTT in media and incubated for 2 h at 37 °C and 5% CO_2_. Afterwards, 50 µL per well of Dimethyl sulfoxide (DMSO) was added to solubilize the crystals. After 30 min at room temperature, the absorbance was measured at a wavelength of 560 nm on a spectrophotometer microplate-reader (BioTek™ Synergy™ 2).

### 2.7. Cell Culture Derived CHIKV Stock Production and Titration 

Plasmid DNA (20 µg) encoding CHIKV genome was linearized using NotI (New England Biolabs) endonuclease restriction digestion. Complete linearization was confirmed by agarose gel electrophoresis and linearized DNA purified using the QIAprep Spin Miniprep kit (QIAgen, Hilden, Germany) following the manufacturer protocol. A 100 µL in vitro transcription reaction was prepared using 2 µg of the DNA mixed with nuclease free water, 10 µL of RNA polymerase buffer (10×), 10 µL of rNTP-mix (25 nM each, Roche, Basel, Switzerland), 5 µL of 5′cap Analog, 2.5 µL of RNAse inhibitor (Promega, 40 U/µL, Madison, WI, USA), and 6 µL of SP6 polymerase (New England BioLabs, Ipswich, MA, USA). The mixture was incubated for 2 h at 37 °C after which 4 µL of SP6 polymerase was added and incubated for a further 2 h. The reaction was stopped by the addition of 7.5 µL of DNAse (Promega, 1 U/µL) and incubated for 30 min at 37 °C to digest the DNA template. The synthesized RNA was purified using NucleoSpin RNA Clean-up kit (Macherey-Nagel, Düren, Germany) and analyzed by agarose gel electrophoresis. Afterwards, the concentration was determined on a spectrophotometer and aliquots of 20 µg frozen at −80 °C. 

To produce the CHIKV reporter viruses, 20 µg of the in vitro transcribed RNA was electroporated into 1 × 10^7^ BHK-21 cells in Opti-Mem^®^ (Gibco™). Electroporation was performed using a Gene Pulser (Bio-Rad, Hercules, CA, USA) at 250 V, two pulses at an interval of one second and 15 ms pulse length. Cells were immediately transferred into 10 mL of complete DMEM and seeded on 10-cm dishes. The supernatant containing CHIKV was collected 48 h post electroporation and cellular debris removed using a 0.45-μm pore size filter. The virus was concentrated by either ultracentrifugation through a 20% sucrose gradient or by use of 100 MW amicon tubes (Merck, Darmstadt, Germany). The supernatant was subsequently stored in small aliquots at −80 °C. Virus titers were assessed by flow cytometry (FACS) and luciferase assay on HEK293T cells and expressed as median Tissue Culture Infectious Dose (TCID_50_).

### 2.8. Infection and Antibody Inhibition Assay

To determine susceptibility to CHIKV, cells were seeded on cell culture plates coated with poly-*L*-lysine (HEK293T) or uncoated (Huh7.5, HaCats, fibroblasts and CHO) at the densities indicated below. After an overnight culture, cells were transferred to biosafety level three lab and incubated with CHIKV at indicated MOI for four hours. The inoculum was replaced with fresh medium and cells incubated for the specified time points. Infection was determined by flow cytometry or luciferase assay depending on the virus used.

For antibody inhibition of CHIKV, HEK293T cells stably expressing TIM-1 WT were seeded in a 96-well plate at a density of 2 × 10^4^ cells/well 24 h prior to the experiment. Cells were preincubated in media with the indicated concentration of anti-TIM-1 polyclonal antibody in DMEM complete. Identical concentrations of IgG isotype were used as control. After 30 min, the cells were inoculated with GFP tagged CHIKV at MOI of 0.01 for 4 h in the presence of the antibody, washed and incubated with culture medium. Susceptibility was analyzed by flow cytometry 24 h post infection.

To determine susceptibility to VEEV, parental HEK293T cells and cells expressing TIM-1 WT or Axl WT were seeded in 96-well plates (2 × 10^4^ cells/well) coated with poly-*L*-lysine. After 24 h, cells were inoculated with virus at MOI of 0.001, 0.01, 0.1, and 1 and incubated for 4 h at 37 °C. The inoculum was removed and fresh DMEM supplemented with 10% FCS added. After 16 h of infection, cells were detached and GFP expression determined by flow cytometry. 

To determine susceptibility to SINV, parental HEK293T cells and cells expressing TIM-1 WT or Axl WT were seeded in 6-well plates (1.3 × 10^6^ cells/well) coated with poly-*L*-lysine. After 24 h, cells were inoculated with virus at MOI of 0.01 and 0.1 and incubated for 1 h at 37 °C. The inoculum was removed and fresh DMEM supplemented with 10% FCS added. The supernatant from the wells was collected after 24 h of infection and titrated at a 10-fold dilution on Vero cells seeded in 12-well plates (5 × 10^5^ cells/well). After 1 h at 37 °C the virus was removed and carboxymethyl cellulose overlay added. After 48 h the wells were fixed with 4% PFA and stained with crystal violet solution. The titer was determined by counting visible plaques.

To determine susceptibility to HAdV-C5-GFP, parental HEK293T cells and cells expressing TIM-1 WT or Axl WT were seeded in black 96-well plates with transparent bottom (3 × 10^4^ cells/well). Cells were washed twice with DMEM before addition of serial dilutions of HAdV-C5-GFP. After 1 h at 37 °C, the virus was removed and fresh DMEM supplemented with 2% FBS was added to the cells. Then, 24 h post infection the plates were fixed with 4% PFA for 10 min and GFP expression was imaged using a Trophos system (Luminy Biotech Enterprises, Marseille, France).

### 2.9. Virus Binding and Endosomal Escape Assays

Cells were suspended in binding buffer (DMEM supplemented with 20 mM HEPES, 1 mM calcium chloride and 0.2% human serum albumin, pH 7.4) with mCherry-fluorescent CHIKV (mc-CHIKV) at MOI of 50 and incubated at 4 °C on a shaker for 2 h. The cells were washed 3 times in binding buffer and fixed using 4% PFA. The cells suspended in FACS buffer were then analyzed for binding by flow cytometry.

For endosomal escape, cells inoculated with nano-luciferase CHIKV (nLuc-CHIKV) in DMEM complete medium were incubated on ice for one hour to allow maximum attachment and synchronized entry of virus. Cells were washed three times with DMEM to remove unbound nLuc-CHIKV. Binding efficiency between parental and TIM-1 WT/mutant expressing HEK293T cells was determined by luciferase assay. After removing unbound virus, fresh DMEM complete was added and cells were transferred to 37 °C to initiate particle uptake. At the indicated time points, medium supplemented with 20 mM ammonium chloride was added in order to prevent endosomal acidification and escape of the virus from endosomes. After 10 h of continuous incubation with ammonium chloride at 37 °C, the cells were lysed by freeze-thawing and productive infection assessed by determining the enzymatic activity of the newly translated luciferase after initial replication of the incoming viral genomes.

### 2.10. Confocal Microscopy and Live Cell Imaging

Live-cell imaging was carried out using a laser confocal spinning-disc coupled to a motorized Ti-E inverted microscope (Nikon, Tokyo, Japan) and equipped with a Yokogawa CSU-X1 5000 Spinning Disk Unit and an EMCCD camera iXon Ultra DU-888 (Andor Technologies, Belfast, Northern Ireland). Time-lapse movies were acquired using 60× and 100× objectives (NA = 1.49) and NIS-Elements AR DUO software and were recorded at an acquisition rate of 3 frames per second for 2 min. 

Prior to imaging, cells were stained using a membrane permeable dye for living cells, Calcein AM (C3099, Invitrogen, Carlsbad, CA, USA) at a concentration of 0.2 μM for 10 min at 37 °C. While Calcein AM is usually used to probe cell viability, here it was used to identify the cells and make it possible to count them using confocal imaging. Cells were kept in a growth chamber (37 °C, 5% CO_2_) for the entire acquisition time. Labeled mc-CHIKV viruses were immediately imaged after they were added to the cells and were excited using the 561 nm laser. Before and after each time-lapse recording, the cells and viruses were imaged using a 488 nm and 561 nm laser excitation to check for autofluorescence. 

Internalized viral particles were quantified by quenching of the extracellular viral particles using Trypan Blue (Gibco). Twenty minutes after addition of mc-CHIKV to the cells, Trypan Blue (Gibco™) was added at the concentration of 0.4% to quench the extracellular viral particles. Zstack images of the cells after addition of Trypan Blue were recorded using a 488 nm and 561 nm laser excitation. The number of internalized particles per cell was then counted using Fiji and the multipoint tool. The number of cells used for the quantification of the bound and the internalized particles was determined using images taken in the green TIM-1 channel. Finally, the total number of virus particles and of intracellular virus particles was each divided by the number of cells.

### 2.11. Single Particle Tracking

Recorded movies of mc-CHIKV diffusion at the cell surface were processed and analyzed using TrackMate [59] and Matlab DC-MSS (Divide-and-Conquer Moment Scaling Spectrum) transient diffusion analysis [60]. First, the movies were pre-processed using Fiji by correcting uneven background using a rolling ball of 50 and by filtering the noise (despeckle). The virus trajectories were then reconstructed using TrackMate (ImageJ, 1.53j, University of Wisconsin, Madison, WI, USA) where the virus particles were detected with sub-pixel localization and linking of frame-to-frame displacement of 1 μm and a maximum gap of 2 μm and 20 frames. Aggregates and large particles were manually excluded from the analysis. Trajectories longer than 60 frames were then segmented and classified in Matlab using a built-in script and DC-MSS. Briefly, diffusion classification was done using the moment scaling spectrum (MSS) where high order moments of the displacement distribution are considered and the slope of the MSS reflects the motion type: a slope of 0.5 implies free normal diffusion, a slope between 0 and 0.5 yields anomalous motion, and a slope of 0 represents immobile particles. Trajectories were segmented depending on the motion type with a rolling-window of 21 frames. For each segment, diffusion properties such as the diffusion coefficient and confinement radius were extracted as detailed by Vega et al. [60]. Moreover, the time spent in immobile, anomalous, or free motion type was calculated by dividing the sum of the time spent in one motion type by the total time spent by all segments in all motion types. 

### 2.12. Western Blot Analysis

Cells were washed three times using PBS and suspended for 30 min on ice in lysis buffer (1% Nonidet P40, 10% glycerol, 1 mM CaCl_2_ in HEPES/NaCl) supplemented with 1% protease inhibitor (Sigma-Aldrich #P8340, Burlington, MA, USA). Supernatants were collected after centrifugation and total protein concentration determined by Bradford assay. Then, 25 µg of total protein was separated in reducing conditions by sodium dodecyl sulphate-polyacrylamide gel electrophoresis (SDS-PAGE). The proteins were transferred to polyvinylidene difluoride (PVDF) membrane (Bio-Rad) followed by blocking with 5% skimmed milk in PBS supplemented with 0.5% Tween 20 (PBS-T). The membranes were incubated with their respective primary antibody for one hour at room temperature. After washing three times using PBS-T, the membranes were incubated with the indicated horse radish peroxidase (HRP)-conjugated secondary antibodies. Following extensive washing, protein levels were detected using ECL Prime Western blot detection system (GE Healthcare, Chicago, IL, USA) and visualized using the ChemoStar Professional Imager System (Intas, Göttingen, Germany).

### 2.13. Luciferase Assay

Luciferase activity was determined as previously described [61] in cells inoculated with lentiviral pseudoparticles or authentic CHIKV and in cells electroporated with CHIKV subgenomic or full-length RNA. Firefly luciferase activity was measured by mixing 20 μL cell lysate with 72 μL firefly luciferase assay buffer [25 mM glycyl-glycine (pH 7.8), 15 mM KPO_4_ (pH 7.8), 15 mM MgSO_4_, 4 mM EGTA, 1 mM DTT and 2 mM ATP (pH 7.6)] and 40 µL of firefly luciferase substrate (0.2 mM D-luciferin in 25 mM glycyl-glycine). Nano-luciferase activity was measured by adding 80 µL of 1:1000 coelenterazine solution (0.42 mg/mL in methanol) to 20 µL of the lysate. Luciferase activity was measured in a plate luminometer (LB960 CentroXS3, Berthold technologies) in white luminometer 96-well plate.

### 2.14. Surface Staining and Flow Cytometry

The expression of TIM-1, Axl, and MXRA8 was analyzed by staining cells with anti-TIM-1, anti-Axl (R&D Systems, Minnneapolis, MN, USA), and anti-MXRA8 (JSR life sciences, Sunnyvale, CA, USA) monoclonal antibodies without prior fixation. The primary staining with unconjugated antibody was followed by secondary staining with either Alexa 488- or 647-conjugated anti-mouse/goat IgG antibody (ThermoFisher, Waltham, MA, USA). The respective IgG isotype was used as control. The expression of TIM-1 and Axl was analyzed by APC-conjugated monoclonal anti-TIM-1 (BioLegend^®^) and anti-Axl (R&D Systems, Minnneapolis, MN, USA) antibodies respectively. Appropriate APC-conjugated isotype control antibodies from BioLegend^®^ and R&D Biosystems were used. All flow cytometry analyses in this study were performed using Sony Spectral Cell Analyzer (Sony Biotechnology, San Jose, CA, USA) and data analyzed by FlowJo V10 Software.

### 2.15. Statistical Analysis

Experiments were performed in at least three biological replicates, each carried out in technical triplicates unless otherwise specified. Results are plotted as mean ± standard error of mean (SEM) of three biological replicates unless otherwise indicated. Statistical analyses were performed in GraphPad Prism 8 (GraphPad Software, Inc., San Diego, CA, USA) using analysis of variance (ANOVA) followed by Dunnett’s multiple comparison test. Statistical relevance for binding and internalization of CHIKV was calculated using Welch t-test. Statistical relevance was reached for *p* ≤ 0.05 (*), *p* ≤ 0.01 (**), *p* ≤ 0.001 (***), and *p* ≤ 0.0001 (****); *p* > 0.05 (ns) was considered non-significant.

## 3. Results

### 3.1. Ectopic TIM-1 Expression Enhances CHIKV Infection in HEK293T Cells

In order to investigate the role of wild type TIM-1 (TIM-1 WT) and Axl (Axl WT) in mediating CHIKV infection, we generated HEK293T cells stably expressing TIM-1 WT and Axl WT (Figure 1A and Appendix A). The immunoblot of TIM-1 (predicted molecular weight = 39.3 kDa) and Axl (predicted molecular weight = 98.3 kDa) confirmed expression of both proteins, however also revealed additional bands at higher molecular weight, which are most likely attributed to post-translational modifications. TIM-1 has several Ser, Thr, and Pro as well as Asn residues in the mucin domain that can be modified by O-linked or N-linked glycans, respectively [62]. Moreover, TIM-1 forms homodimers due to high affinity between residues in the IgV domains [63,64]. Axl mainly undergoes N-linked glycosylation [65]. The predicted Axl molecular weight of ~98.3 kDa may explain the lower band. However, the band with molecular weight of ~130 kDa is considered to be fully glycosylated and functional [66]. We inoculated cells with increasing multiplicities of infection (MOI) and analyzed susceptibility to the green fluorescent protein encoding CHIKV (3′GFP-CHIKV) by flow cytometry after 24 h. The cells that ectopically express TIM-1 WT were more susceptible to CHIKV, especially at MOI ≤ 0.1. At MOI of 0.01, TIM-1 WT expression increased susceptibility by 12-fold whereas Axl WT expression resulted in a two-fold increase (Figure 1B,C). Notably, TIM-1 WT and Axl WT expressing HEK293T cells were more susceptible to CHIKV (33-fold and 10-fold respectively) and Ebola virus (18-fold and five-fold respectively) glycoprotein-based lentiviral pseudoparticles demonstrating that the proteins are functional as previously reported [18,67,68,69] (Appendix A). Next, we analyzed the susceptibility of the cells to authentic CHIKV at different hours post infection (hpi). At 24 hpi, TIM-1 WT expressing cells were 69% positive for CHIKV while parental cells and Axl WT expressing cells were 11% and 21% respectively positive for CHIKV (Figure 1C). To evaluate the dependence of different CHIKV genotypes on TIM-1 WT, we inoculated cells with strains of ECSA (MOI = 0.01), West African (WA, MOI = 0.01), and Asian (181/25, MOI = 0.1) genotypes. In comparison to the control cells, TIM-1 WT expression in HEK293T cells consistently enhanced the infection with all tested CHIKV strains (ECSA: 28-fold, WA: 14-fold, and 181/25: 10-fold) (Figure 1D). Moreover, we observed a dose-dependent inhibition of 3′GFP-CHIKV infection of TIM-1 expressing cells using an anti-TIM-1 polyclonal antibody (α-TIM-1 Ab) while an isotype IgG control antibody slightly affected CHIKV infection, but not in a dose-dependent manner (Figure 1E). 

In order to establish the role of TIM-1 in the presence or absence of glycosaminoglycans (GAGs), we expressed human TIM-1 in Chinese hamster ovary (CHO) cells with (CHOK1) and without GAGs (CHO745) [70,71]. We inoculated CHO cells with strains of ECSA (MOI = 0.01), WA (MOI = 0.01), and Asian (MOI = 0.1) genotypes. In comparison to control cells, the expression of TIM-1 in CHOK1 cells increased susceptibility by 11-fold (ECSA), 21-fold (WA), and 15-fold (181/25) whereas a ~7-fold increase was observed in CHO745 for all tested genotypes. CHO745 cells expressing TIM-1 were generally approximately two-fold less susceptible to CHIKV than CHOK1 expressing TIM-1 (Appendix A) despite similar susceptibility of parental cells lacking TIM-1 (Appendix A). This suggests that the observed TIM-1-dependent enhancement of CHIKV infection is slightly modulated by the expression of GAGs. 

To test if other alphaviruses use TIM-1 and/or Axl as host factor, we inoculated HEK293T cells expressing TIM-1 and Axl with Venezuelan eastern equine encephalitis virus (VEEV) and with Sindbis virus (SINV) at increasing MOI. In comparison to the parental cells, TIM-1 expression enhanced VEEV infection by 1.4-fold at a MOI of 0.1 (Appendix A). SINV infection increased four-fold upon TIM-1 expression as measured by infectious particle release (Appendix A). As a control, we challenged the cells with serially diluted human adenovirus-5 (HAdV-5), a non-enveloped virus. Infection of the cells was independent of TIM-1 and Axl (Appendix A). Taken together, these observations indicate that unlike Axl, TIM-1 expression enhances infection of different CHIKV genotypes and alphaviruses. However, in comparison to CHIKV, VEEV and SINV appear to be less dependent on TIM-1.

In order to establish if TIM-1 has a role in the replication of CHIKV, we generated HEK293T cells expressing TIM-1 wild type (TIM-1 WT), TIM-1 with a double mutation in the PtdSer-binding pocket (TIM-1ΔMIL, N114A, and D115A), and TIM-1 lacking the cytoplasmic domain (TIM-1ΔCyt). We then compared the replication in parental cells to TIM-1 WT, TIM-1∆MIL, and TIM-1∆Cyt by electroporating CHIKV subgenomic RNA encoding for nano-luciferase. Our findings showed similar RNA replication between the cells as determined by luciferase assay (Appendix A). Furthermore, we electroporated the cells with the full length CHIKV RNA encoding nano-luciferase gene (nLuc-CHIKV) to confirm the observation made by electroporating subgenomic RNA. Infection was stopped at 4, 6, 8, and 10 h timepoints to avoid re-infection by de novo virus. By use of the luciferase assay, we observed luciferase activity across the cell variants (Appendix A), suggesting that TIM-1 has no influence on the replication of CHIKV in HEK293T cells. Collectively, this finding implies that the role of TIM-1 in CHIKV infection may be at the level of binding and entry. 

### 3.2. The TIM-1 Phosphatidylserine-Binding Domain Is Crucial for TIM-1-Dependent Infection

Next, we characterized the role of intra- and extracellular domains of TIM-1 in enhancing CHIKV binding, uptake, and infection using HEK293T cells expressing TIM-1 WT, TIM-1ΔMIL, and TIM-1ΔCyt (Figure 2A). We stained TIM-1 WT and TIM-1 deletion mutant expressing cells with antibodies against the ectodomain and sorted cell populations with similar TIM-1 surface expression levels (mean fluorescent intensity, MFI) (Figure 2B). We challenged the parental HEK293T cells and cells expressing TIM-1 WT, TIM-1ΔMIL, or TIM-1ΔCyt with ECSA (MOI 0.01), WA (MOI 0.01), and Asian (MOI 0.1) strains of CHIKV. After 24 h, infection levels in cells expressing the ectodomain mutant (TIM-1∆MIL) were similar to those of parental cells (Figure 2C). Conversely, we observed increased infection levels in cells expressing TIM-1 WT and TIM-1∆Cyt. Expression of TIM-1 WT increased infection with ECSA, WA and Asian (181/25) strains by 25-fold, 21-fold and five-fold, respectively while expression of TIM-1ΔCyt increased infection by 21-fold, 17-fold and seven-fold, respectively (Figure 2C). Notably, infectivity of the Asian vaccine strain (181/25) was lower in comparison to ECSA and WA stains. The lower infectivity is attributed to attenuation due to substitution of two amino acids at positions 12 and 82 in the E2 envelope glycoprotein responsible for receptor binding [52,72]. We observed similar infectivity between cells expressing TIM-1 WT and TIM-1∆Cyt, implying that the cytoplasmic domain of TIM-1 is dispensable for CHIKV infection of HEK293T cells. To further characterize the role of TIM-1 ectodomain in CHIKV infection, we determined the competence of parental cells and cells expressing TIM-1 WT or TIM-1ΔMIL to bind and internalize mCherry-fluorescent CHIKV (mc-CHIKV, Asian genotype). The mc-CHIKV presents the mCherry on the virion surface due to a mCherry-E2 fusion and can be detected by flow cytometry or confocal microscopy. After two hours of CHIKV binding on ice to avoid internalization, we assessed cell bound virus particles by flow cytometry. In comparison to the parental cells, the expression of TIM-1 WT or TIM-1ΔMIL resulted in increased binding of CHIKV (Figure 2D). This increase was more pronounced for TIM-1 WT than for the ectodomain mutant, however this observation did not reach statistical significance in this assay. Imaging of CHIKV binding within the first 20 min of a live cell confocal experiment at 37 °C further confirmed that cells expressing TIM-1 WT bind 2-fold more CHIKV particles compared to TIM-1ΔMIL and this observation reached statistical significance (Figure 2E). Collectively, our findings here indicate that CHIKV depends on the phosphatidylserine binding domain of TIM-1 for efficient binding and infection.

### 3.3. Single Particle Tracking of CHIKV Confirms PtdSer Domain Requirement

To characterize the effect of the PtdSer binding site on the diffusive behavior of CHIKV on the cell membrane, we used single particle tracking of live cells. The mc-CHIKV particles were added to the cells immediately prior to imaging and movies were recorded at three frames per second for two minutes. A total of N_virus_ = 1523 and 472 virus particles for TIM-1 WT and TIM-1ΔMIL respectively were analyzed and a representative trajectory of CHIKV on TIM-1 WT cells is shown in Figure 3A, illustrating the lateral diffusion of the virus at the cell surface (movies. SA-B). The extracted trajectories were further analyzed by segmenting each track depending on the three-diffusion types: immobile and the mobile motions, anomalous confined and free motion (Figure 3B,C corresponding to the track shown in Figure 3A). The average segment length ranged between 12 and 30 seconds with the shortest being six seconds. While in all cases the virus spent 8% and 7% (for WT and mutant respectively) of the time being immobile, for mobile particles, the characteristics of virus diffusion at the cell surface were affected by the mutation in the PtdSer binding site. Indeed, upon mutation of the TIM-1 MILIBS, CHIKV spent less time diffusing freely (35% vs. 48% for the TIM-1 WT cells) (Figure 3D), albeit with a higher diffusion coefficient (Figure 3E). The confinement radius also increased slightly (Figure 3F), in spite of the fact that the anomalous diffusion coefficient remained unaffected. Together, these results reveal that the PtdSer binding site of TIM-1 contributes to modulating the lateral virus Brownian movement at the plasma membrane by reducing the diffusion coefficient of the free motion and the area of virus diffusion at the cell surface, leading to a more confined motion, which could benefit virus internalization.

### 3.4. Entry Kinetics of CHIKV Are Altered by TIM-1

Next, we aimed to dissect the role of TIM-1 protein domains in the initial steps of the CHIKV infection cycle. After binding to the cell surface, CHIKV is thought to be primarily internalized by clathrin-mediated endocytosis and reach endosomes where low pH-dependent membrane fusion occurs and the nucleocapsid is released (endosomal escape) [40,41]. Eventually, the capsid dissociates and the released genome undergoes translation and replication [9]. Addition of medium supplemented with 20 mM of lysosomotropic ammonium chloride (NH_4_Cl) raises the pH in the endosomes hence blocking the endosomal escape of viruses [73,74,75]. To that end, we compared the CHIKV endosomal escape kinetics in HEK293T cells expressing either TIM-1 WT, TIM-1ΔMIL, or TIM-1ΔCyt.

We inoculated cells with CHIKV carrying a nano-luciferase fused E2 glycoprotein (nLuc-CHIKV) and synchronized virus binding for one hour in the cold. After extensive washes, we determined binding efficiency by luciferase assay, making use of the virion incorporated nano-luciferase E2 fusion protein (Appendix A). To initiate CHIKV internalization and endosomal escape, we then shifted the cells to 37 °C. At the indicated timepoints, we exposed the cells to 20 mM NH_4_Cl to raise the endosomal pH and prevent membrane fusion. After 10 h at 37 °C, we quantified the enzymatic activity of the luciferase E2 fusion protein translated from newly replicated CHIKV genomes as a measure for productive infection (Figure 4A). In comparison to the parental cells, TIM-1 WT and TIM-1ΔCyt expressing cells more efficiently bound nLuc-CHIKV by three-fold (Appendix A). To determine TIM-1-dependent entry/endosomal escape kinetics, and to exclude the role of other cellular factors, we normalized the data to that of the parental cells at each time point. We observed a TIM-1 dependent enhancement of entry kinetics after 20 min and for all subsequent timepoints. We also observed enhanced endosomal escape as compared to parental cells in cells expressing TIM-1ΔCyt and this became apparent at one hour post temperature shift. In contrast, cells expressing TIM-1ΔMIL showed no significant enhancement of endosomal escape as compared to parental cells (Figure 4B and Appendix A). Thus, although cells expressing either TIM-1 WT or TIM-1ΔCyt bound equal numbers of CHIKV particles (Appendix A), TIM-1ΔCyt expressing cells needed more time to internalize CHIKV in comparison to TIM-1 WT expressing cells. Similarly, after 20 min of incubation at 37 °C, we observed by confocal microscopy that the number of internalized mc-CHIKV virions in TIM-1 WT expressing cells was three-fold higher than the number of virions in cells expressing TIM-1ΔMIL (Figure 4C). Altogether, these results indicate that, in addition to the TIM-1 PtdSer binding domain, the cytoplasmic domain may modulate TIM-1-dependent CHIKV entry kinetics.

### 3.5. TIM-1 Expression Renders Keratinocyte Derived HaCat Cells Permissive to CHIKV

The skin is the primary entry point of CHIKV. Human epidermal keratinocytes express Axl and TIM-1 in the stratum basale layer [37,38] and are susceptible to CHIKV [20]. In order to determine the role of Axl and TIM-1 in CHIKV infection of the skin, we used HaCat cells, a derivative of immortalized keratinocytes that acts as a relevant model to study keratinocytes in vitro [39]. However, surface staining using monoclonal antibodies and flow cytometry revealed that HaCat cells endogenously express modest levels of Axl (Figure 5A), but no detectable levels of TIM-1 (Figure 5B). Hence, we used lentiviral pseudoparticles to generate cells stably expressing Axl WT, Gas6 binding site mutant (Axl E66R_T84R), tyrosine kinase domain mutant (Axl K567A), Axl lacking the cytoplasmic domain (AxlΔCyt), and Axl with a naturally occurring single nucleotide polymorphism (SNP, Axl R295W) (Figure 5A and Appendix A). We also generated cells expressing TIM-1 wild type (TIM-1 WT), TIM-1 with a mutation in the PtdSer-binding pocket (TIM-1ΔMIL including N114A and D115A), TIM-1 lacking the cytoplasmic domain (TIM-1ΔCyt), TIM-1 with single and double mutations (K338A, K346A, and TIM-1ΔUbi) in the cytoplasmic ubiquitination motif, TIM-1 lacking the cytoplasmic domain (TIM-1ΔCyt), and TIM-1 with a naturally occurring single nucleotide polymorphism (SNP) in the Ig-V domain (TIM-1 S51L) (Figure 5B and Appendix A). We detected expression of all Axl and TIM-1 variants on the surface of HaCat cells. Expression levels were comparable with the exception of the Axl E66R_T84R and the TIM-1-1ΔCyt, which displayed slightly reduced expression as compared to the respective WT protein.

In order to establish if Axl and TIM-1 have a role in CHIKV entry into HaCat cells, we transduced the cells with lentiviral pseudoparticles (pp) decorated with glycoproteins of CHIKV (CHIKVpp) and VSV (VSVGpp). In comparison to parental HaCat cells, which did not support CHIKV pseudoparticle entry, Axl WT enhanced CHIKVpp and VSVGpp entry into HaCat cells by ~two-fold while all the Axl mutant variants did not enhance entry (Appendix A). This was in contrast to the ten-fold CHIKVpp increase in entry observed in HEK293T cells (Appendix A). TIM-1 WT expression enhanced pp entry (five-fold and nine-fold for CHIKVpp and VSVGpp, respectively), as did the TIM-1 mutants TIM-1ΔUbi, TIM-1 K338R and TIM-1 K346R (five-fold and six-fold for CHIKVpp and VSVpp respectively), TIM-1ΔCyt (two-fold for both CHIKVpp and VSVGpp) and TIM-1 S51L (three-fold and five-fold for CHIKVpp and VSVGpp respectively). There was no entry enhancement of CHIKVpp or VSVGpp entry into cells expressing the PtdSer-binding pocket mutants; TIM-1ΔMIL, N114A and D115A (Appendix A). Unlike expression in HEK293T cells, TIM-1 WT expression in HaCat cells enhanced VSVGpp entry implying that VSV dependence of TIM-1 is cell-type specific. Our results suggest that the ubiquitination motif of TIM-1 is dispensable for entry into HaCat cells whereas the cytoplasmic domain in general, the PtdSer-binding pocket and the naturally occurring ectodomain SNP (TIM-1 S51L) are required for the full CHIKV entry factor function of TIM-1 in HaCat cells.

We next challenged the Axl and TIM-1 expressing HaCat cells with authentic nLuc-CHIKV and determined infectivity by luciferase assay. In comparison to the parental cells, which were refractory to CHIKV infection, HaCat cells expressing TIM-1 WT were 14-fold and 22-fold more permissive to nLuc-CHIKV at 24 hpi and at 48 hpi, respectively (Figure 5C and Appendix A). Interestingly, neither the expression of TIM-1ΔMIL, TIM-1ΔCyt, nor Axl rendered HaCat cells susceptible. In contrast to HEK293T cells, where the cytoplasmic domain of TIM-1 was dispensable for CHIKV infection (Figure 2C), the cytoplasmic domain of TIM-1 was required for infection of HaCat cells (Figure 5C and Appendix A). In order to establish if infection of HaCat cells was productive, we collected HaCat cell supernatants at the indicated time points post infection, inoculated human dermal fibroblasts with the supernatants for 24 h and determined CHIKV infection of the fibroblasts by luciferase assay. Fibroblasts are known to be permissive to CHIKV and the infection is MXRA8-dependent [20], however our data cannot exclude a role for Axl in this cell line. In comparison to the parental cells, expression of TIM-1 WT, resulted in a 10-fold and 24-fold higher release of infectious CHIKV particles at 24 hpi and 48 hpi, respectively. Expression of TIM-1ΔCyt and TIM-1ΔMIL in HaCat cells resulted in a five-fold and two-fold higher release of CHIKV compared to parental cells. The expression of Axl did not yield infectious CHIKV particle release (Figure 5D). Altogether, these findings underpin that both the ectodomain and the cytoplasmic domain of TIM-1 complement each other and play a role in CHIKV infection. The findings also demonstrate the cell type specific dependence of CHIKV on TIM-1.

### 3.6. Endogenous TIM-1 Mediates CHIKV Infection of Hepatoma Cells

Chikungunya virus has a wide tissue and cellular tropism and previous reports indicate that it infects the liver [76,77]. Hence, we determined the expression levels of TIM-1 and Axl in Huh7.5 cells—a human hepatoma derived cell line. We found that Huh7.5 cells predominantly express TIM-1 while Axl expression is negligible (Figure 6A and Appendix A). To analyze the role of TIM-1 in CHIKV infection of Huh7.5 cells, we used a pool of three siRNAs to specifically knockdown TIM-1, MXRA8, a known factor for CHIKV entry (as positive control) [20,22], or both (Appendix A). The viability of cells transfected with a pool of siRNA was similar to control cells transfected with a non-targeting (NT) siRNA (Figure 6B). 48 h post silencing, the surface expression (mean fluorescent intensity, MFI) of TIM-1 and MXRA8 was reduced by four-fold and two-fold, respectively (Figure 6C). In comparison to cells transfected with NT siRNA, we observed that siRNA mediated silencing of TIM-1 reduced susceptibility to CHIKV by ~25%. Knockdown of MXRA8 resulted in ~60% reduction in susceptibility to CHIKV similar to simultaneous silencing of TIM-1 and MXRA8 (Figure 6D). Together, these findings show that, in the presence of MXRA8, endogenous TIM-1 plays a role in CHIKV infection of Huh7.5 cells.

## 4. Discussion

Phosphatidylserine-binding proteins, including TIM-1 and Axl receptor tyrosine kinase, are important host factors for a number of viruses [18,33,34,78,79,80]. In the present study, we demonstrated that human TIM-1 plays a key role in CHIKV infection of human cells. In contrast to findings in previous studies that used CHIKV glycoprotein pseudotyped viruses, we observed negligible enhancement of susceptibility to authentic CHIKV in cells expressing Axl [19,80]. TIM-1 associates with PtdSer on virions through the immunoglobulin-like variable (Ig-V) domain whereas Axl interacts through a growth arrest-specific 6 ligand (Gas6) or protein-S to enhance viral entry, which may explain the differential use of both proteins observed in this study [18,33,34,79].

HEK293T cells expressing TIM-1 exhibited a dose-dependent reduction in susceptibility to CHIKV when pre-treated with a TIM-1 specific antibody. This is an indication that the ectodomain of TIM-1 is critical for its function in CHIKV infection. Furthermore, ectopic expression of TIM-1 in HEK293T cells enhanced binding and subsequent internalization of CHIKV particles. The PtdSer-binding pocket also known as metal ion ligand-binding site (MILIBS) is conserved across all TIMs [81]. We mutated the MILIBS in human TIM-1 by replacing Asn and Asp residues (N114A and D115A) or both (TIM-1ΔMIL) with Ala at positions 114 and 115 respectively. We observed that the HEK293T cells expressing TIM-1ΔMIL similarly bind CHIKV as cells expressing TIM-1 WT when incubated with the mCherry-fluorescent virus on ice and detected by flow cytometry. This suggests that apart from the MILIBS, other residues within TIM-1 or other factors on the cell surface contribute to CHIKV binding [82]. However, when CHIKV was added to the cells in normal medium and immediately observed by confocal microscopy at 37 °C, CHIKV bound more efficiently to cells expressing TIM-1 WT in comparison to TIM-1ΔMIL. Consistently, we observed that subsequent internalization and infection was completely hampered in cells expressing TIM-1ΔMIL. According to our findings, the residues in the MILIBS are essential for TIM-1-mediated CHIKV infection and this is in line with previous reports [18,33,34,80]. Ectodomain residues outside the MILIBS were found to be less important in Dengue virus (DENV) infection [34]. Conversely, EBOV infection is additionally mediated by the direct interaction between viral glycoprotein (GP) and TIM-1 ectodomain residues outside the MILIBS but within the Ig-V domain [34,83,84]. Our study shows that TIM-1 residues in the Ig-V domain other than in the MILIBS are also needed for the entry factor function of TIM-1 in the context of CHIKV infection. Specifically, we observed reduced CHIKV pseudoparticle entry in HaCat cells expressing TIM-1-S51L in comparison to cells expressing wild type TIM-1. This observation suggests that TIM-1 residues outside the MILIBS may play a role in CHIKV infection implying that different viruses use a distinct set of TIM-1 residues for infection. A better understanding of the molecular interaction between amino acid residues in the IgV domain of TIM-1, PtdSer and viral glycoproteins may help in the development of antiviral factors. For instance, Song et al. recently developed a reagent that specifically binds PtdSer and/or phosphatidylethanolamine and could inhibit ZIKV infection [85]. Analysis of the diffusive behavior of CHIKV upon binding to the cell surface not only confirms that the PtdSer binding site promotes virus binding but also influences its diffusive behavior. Specifically, expression of TIM-1 with an intact PtdSer binding site leads to a decrease in the diffusion coefficient of the virus as well as the area of diffusion, indicating that the virus binds to TIM-1 WT directly or indirectly within membrane protein complexes. CHIKV diffusion may be slowed down by assembly of a higher molecular weight complex. This is in line with a TIRF microscopy study which reported that 76% of TIM-1-GFP spots on HeLa cells are confined, 19% transported, and 5% diffusive [37]. The majority of TIM-1-GFP tracked for Dengue virus internalization displayed confined displacement at the plasma membrane [37]. As CHIKV diffusion coefficient and confinement radius increased for TIM-1ΔMIL expressing cells, one can hypothesize that the PtdSer binding site stabilizes the interaction of CHIKV with TIM-1 protein complexes. This further highlights the role of the PtdSer binding site in the attachment of CHIKV to the cell membrane. 

The hampered entry and infection in cells expressing TIM-1ΔMIL may signify that the MILIBS is involved in signaling for internalization of CHIKV, possibly by mediating interaction with signaling receptors. Our data suggest that TIM-1 signaling through the cytoplasmic tail is dispensable for CHIKV entry into HEK293T cells as the cytoplasmic tail deletion of TIM-1 (TIM-1ΔCyt) does not impact CHIKV infection at a later time point. A similar observation was seen in DENV infection of HEK293T cells expressing TIM-1ΔCyt [34]. However, the endosomal escape assay showed that CHIKV entry into cells expressing TIM-1ΔCyt was slower in comparison to TIM-1 WT cells, suggesting that the cytoplasmic domain of TIM-1 may support efficient internalization. The importance of the cytoplasmic domain in TIM-1-dependent CHIKV infection was apparent in HaCat cells expressing TIM-1ΔCyt. Here, only TIM-1 WT enhanced CHIKV infection while cells expressing TIM-1ΔCyt remained refractory to CHIKV infection similar to parental HaCat cells. The two lysine residues at positions K-338 and K-346 in the cytoplasmic domain of TIM-1 are targets of ubiquitin ligases [37]. Since ubiquitin chains are internalization signals, the lysine residues have the potential to initiate internalization of TIM-1 upon ubiquitination [86]. In this study, HaCat cells expressing TIM-1 ubiquitination motif mutants were still susceptible to CHIKV pseudoparticles implying that CHIKV infection of HaCat cells is independent of TIM-1 ubiquitination. Taken together, these findings indicate that TIM-1 mediates CHIKV infection by enhancing particle attachment and uptake into cells. Our data further show that the role of the ectodomain and cytoplasmic domain of TIM-1 may be cell type specific and presumably depends on the presence or absence of other attachment factors.

Previous studies on cell entry of alphaviruses have pointed out different entry pathways, including macropinocytosis [87] as well as clathrin-dependent and -independent endocytosis [8,42,88,89], which suggests cell type specific variations. Application of the lysosomotropic agent ammonium chloride led to reduced CHIKV infectivity confirming that TIM-1-mediated entry of CHIKV occurs via an endocytic pathway. Since viruses are obligate intracellular pathogens, fast cell entry benefits the maintenance of virus structural integrity for effective intra-cellular delivery of its genomic material. Delayed entry may lead to virus inactivation in the extracellular milieu for instance due to variations in pH [90]. The endosomal escape assay further emphasized the importance of the MILIBS and cytoplasmic domain of TIM-1 in rapid CHIKV entry into cells. 

Li et al. observed that expression of TIM-1 inhibits HIV-1 release due to the association of PtdSer-binding domain with the PtdSer on the membranes of the budding virions leading to diminished virus production and replication [91]. Interestingly, TIM-1 was shown to increase replication and virus production of Japanese encephalitis virus (JEV) [32]. In the current study, we did not observe an increase in CHIKV replication upon expression of TIM-1 in HEK293T cells. Moreover, infected HaCat cells expressing TIM-1 WT produced more infectious CHIKV particles, likely due to enhanced initial virus entry. These results argue that, unlike for HIV-1, the enhanced entry of CHIKV due to the expression of TIM-1 WT is dominant over possible inhibition during release, resulting in increased viral production. Additionally, TIM-1-dependent inhibition of HIV-1 release may be attributed to the lower density of envelope glycoproteins in comparison to other viruses [92], hence PtdSer in the budding virus particles is readily accessible for binding TIM-1. Altogether, these observations suggest that TIM-1 may have different roles in specific virus families and these may yet be cell type dependent.

In the skin, which is the primary entry point of the mosquito transmitted CHIKV, TIM-1 and Axl are expressed by human epidermal keratinocytes [37,38]. HaCat cells derived from spontaneously immortalized keratinocytes act as a relevant model to study keratinocytes in vitro [39]. However, HaCat cells endogenously express Axl but not TIM-1 and they are refractory to CHIKV [93]. In our gain-of-function study, the ectopic expression of TIM-1 WT rendered HaCat cells permissive to authentic CHIKV. Conversely, expression of TIM-1ΔMIL, TIM-1ΔCyt and Axl in HaCat cells did not support CHIKV infection. Bernard et al. demonstrated that HaCat cells are refractory to CHIKV due to induction of interferon [93]. However, in the presence of TIM-1 WT we observed increased CHIKV susceptibility and premissiveness. Since the basal layer of epidermal keratinocytes expresses TIM-1 and HaCat cells become permissive to CHIKV upon ectopic expression of TIM-1, TIM-1 may play a physiological role in CHIKV infection of the skin. However, future experiments using primary keratinocytes will need to test this hypothesis. Our findings emphasize the role of the PtdSer binding domain and the cytoplasmic domain of TIM-1 in HaCat cells and demonstrate a cell type specific dependency of CHIKV on TIM-1. The requirement for the cytoplasmic domain of TIM-1 in CHIKV infection was more evident in HaCat cells, which remained non-permissive unlike HEK293T cells. This observation may be attributed to presence of alternative attachment factors in HEK293T cells, which render them readily susceptible even in the absence of TIM-1. 

In order to analyze the role of endogenously expressed TIM-1 in CHIKV infection, we used the hepatoma derived Huh7.5 cell line [49], physiologically relevant cells since the virus infects the liver [76]. Huh7.5 cells predominantly express TIM-1 while Axl expression is negligible. Additionally, Huh7.5 cells express MXRA8, a known entry factor for CHIKV and other alphaviruses [20]. TIM-1 was shown to support binding and infection of hepatitis C virus (HCV) in Huh7.5 cells [94]. In a TIM-1 loss-of-function study, we silenced TIM-1 and MXRA8 and observed a reduction in CHIKV susceptibility for single and double knock downs. This suggests that TIM-1 plays a role in CHIKV infection in the presence of MXRA8. The observed residual susceptibility may be due to incomplete knock down. Additionally, hepatocytes are known to express TIM-1 splice-forms, which lack cytoplasmic tyrosine phosphorylation motif [95]. Hence, TIM-1 entry factor function may be partially impaired in these cells. Our results corroborate the observation made by Jemielity et al., that antibody blocking of TIM-1 barely inhibits entry of CHIKV pseudovirus in Huh7 cells [33]. Overall, these observations indicate that TIM-1 is not the only internalization factor in hepatocytes, however it may serve to concentrate CHIKV on the plasma membrane for subsequent internalization by alternative cellular factors. Glycosaminoglycans (GAGs) may act as co-receptors and additionally concentrate the virus [96]. Here, we observed that the expression of TIM-1 in cells with and without GAGs resulted in enhanced CHIKV infection, although GAGs led to an additional slight increase in susceptibility to CHIKV. This finding further suggests that CHIKV uses a number of cellular factors to broaden its tropism.

Wang et al. used CRISPR/Cas9-mediated gene editing to knockout TIM-1 in Huh7 cells and demonstrated that HCV genome replication was not dependent on expression of TIM-1 [94]. In this study, we investigated the role of TIM-1 in CHIKV replication by electroporating CHIKV subgenomic replicon RNA and full length CHIKV RNA into HEK293T cells expressing wild type TIM-1 and mutant variants. We found that expressing TIM-1 WT did not alter CHIKV replication compared to parental cells. This suggests that TIM-1 does not influence CHIKV genome replication but instead facilitates binding of viral particles on the cell surface to promote subsequent uptake and infection.

Axl is thought to enhance infection of Zika virus by antagonizing immune response [97,98]. In the current study, we did not observe a significant Axl-dependent enhancement of authentic CHIKV infection. HaCat cells which moderately express Axl were refractory to CHIKV infection even after transducing the cells to ectopically express more Axl. Ectopic expression of mutant Axl in HaCat cells served as additional control since Axl is known to dimerize upon activation and the mutant acts as a dominant negative [99]. The presence of mutant forms of Axl would disrupt the activity of the dimers formed and reduce the function of endogenous Axl. In line with the findings for WT Axl, we observed no reduction of infection with mutant Axl. We believe that the ectopically expressed WT Axl was functional as it enhanced the entry of CHIKV and Ebola virus glycoprotein-based lentiviral pseudoparticles. We speculate that the enhanced pseudovirus entry is due to exposure of PtdSer by lentivirus-based CHIKV pseudoparticles and hence binding of the particles to either Axl or TIM-1. The glycoproteins on lentiviral pseudoparticles are likely less densely packed than on authentic alphaviruses and hence PtdSer may be better accessible [92]. Hence, Axl may not efficiently bind authentic CHIKV membrane lipids through its ligands Gas6. Interestingly, mutating the TIM-1 PtdSer binding site abrogated CHIKV infection implying that either TIM-1 unlike Axl is able to access PtdSer in CHIKV particles for binding or that the TIM-1 MILIBS is required for a secondary function of TIM-1 necessary for its role as entry factor. Additionally, based on our observation that the TIM-1 ectodomain residue S51, which is outside the MILIBS, is required for CHIKV entry and a previous findings that TIM-1 interacts with EBOV glycoproteins [84], we speculate that in addition to PtdSer, CHIKV glycoproteins may interact with TIM-1. However, future experiments are required for clarification. Overall, the role of Axl as a virus cell entry factor is inconclusive as conflicting results have been published [75,97,100,101]. We postulate that authentic CHIKV infection is less dependent on Axl. However, additional studies on cells with relevant endogenous Axl expression are needed to confirm this notion. 

Like other alphaviruses, CHIKV induces apoptosis [102,103,104], which is associated with PdtSer exposure on the outer leaflet of the plasma membrane where budding of virions occurs [105,106]. Consequently [107], CHIKV particles may acquire an apoptotic bleb-like membrane during egress and become a target for cells that express TIM-1 or other proteins of the TIM family. Apart from epithelial cells and T-helper cells (Th2), macrophages express TIM proteins which interact with PdtSer in the process of clearing apoptotic bodies [81]. The mouse ortholog of TIM-1 is preferentially expressed in Th2 cells and modulates T-cell activation and proliferation by signal transduction, increasing airway inflammation and allergy [108,109]. The extent to which TIM-1-mediated uptake of CHIKV into immune cells plays a role in antiviral immune responses requires further investigation.

In conclusion, our findings show that TIM-1 enhances CHIKV cell binding and entry, which may have implications for virus propagation and spread. The role of TIM-1 and its domains is cell line dependent and since it is endogenously expressed by primary keratinocytes, HaCat cells that ectopically express TIM-1 could act as a suitable model system. Ultimately, a better understanding of the interaction of CHIKV and cellular factors such as TIM-1 may inform the development of antiviral strategies to combat chikungunya fever.

## Figures and Tables

**Figure 1 cells-10-01828-f001:**
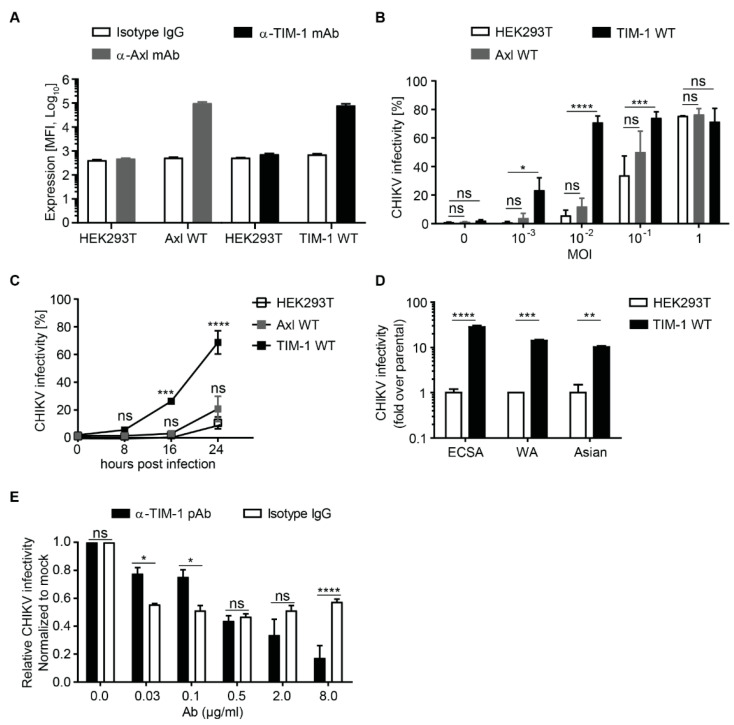
Ectopic expression of TIM-1 enhances CHIKV infection (**A**) Surface expression of Axl and TIM-1 wild type (WT) proteins on parental HEK293T and stably transduced cells was evaluated by monoclonal antibody staining and flow cytometry. Cells stained with an isotype IgG were used as control. (**B**) Parental, Axl WT and TIM-1 WT expressing cells were challenged with ECSA 3′GFP-CHIKV at indicated MOI and (**C**) for different infection durations at MOI of 0.01. Infection levels were assessed by flow cytometry and plotted as percentage of GFP positive cells. (**D**) Parental and TIM-1 WT expressing cells were challenged with CHIKV strains of ECSA 3′GFP-CHIKV (MOI = 0.01), WA 5′GFP-CHIKV (MOI = 0.01) and Asian mc-CHIKV (MOI = 0.1) genotypes and infection assessed by flow cytometry. (**E**) HEK293T cells expressing TIM-1 were pre-incubated for 30 min with increasing concentrations of TIM-1 polyclonal antibody (black bars) or isotype control antibody (mock, white bars) before inoculation with ECSA 3′GFP-CHIKV at MOI of 0.01. After 4 h the cells were washed and infection levels analyzed by flow cytometry 20 h later as in (**B**) and (**C**). Error bars represent standard error of the mean (SEM) of three biological replicates. Statistical significance was calculated using a Dunnet’s multiple comparisons test (2way ANOVA) ns > 0.05, * *p* < 0.05, ** *p* < 0.01, *** *p* < 0.001 and **** *p* < 0.0001.

**Figure 2 cells-10-01828-f002:**
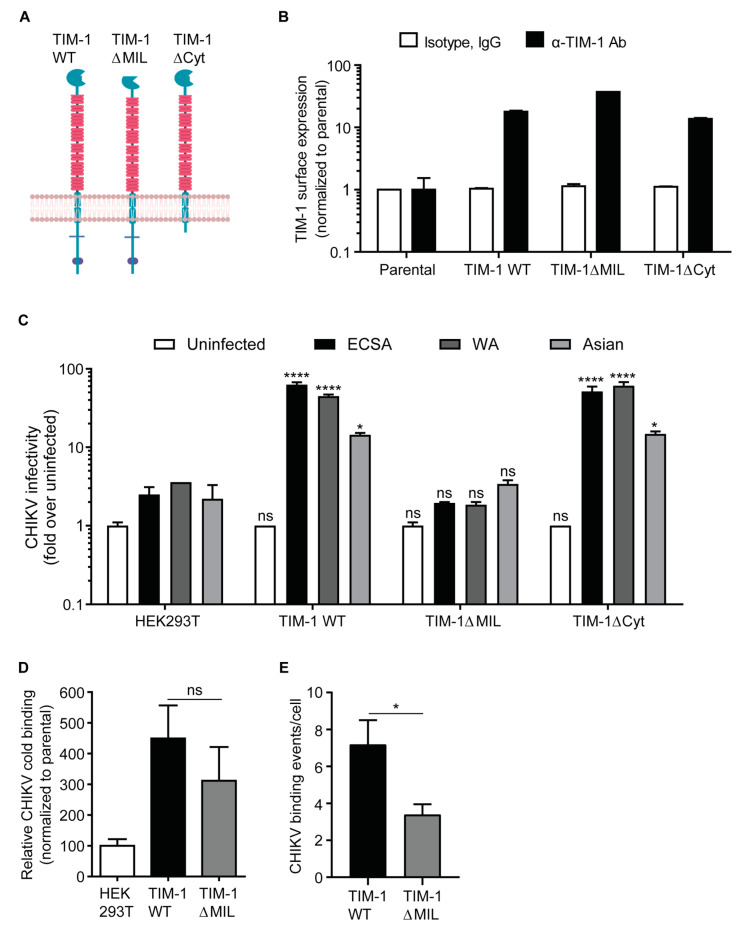
Phosphatidylserine-binding domain of TIM-1 is crucial for CHIKV entry and infection (**A**) Schematic representation of TIM-1 WT, TIM-1ΔMIL with a mutation in the phosphatidylserine binding site and TIM-1ΔCyt that lacks part of the cytoplasmic domain (made with BioRender.com). (**B**) Expression levels of TIM-1 in parental and HEK293T cells transduced with lentiviral pseudoparticles to stably express TIM-1 WT, TIM-1ΔMIL or TIM-1ΔCyt assessed as in Figure 1. (**C**) Parental HEK293T cells expressing TIM-1 WT, TIM-1∆MIL and TIM-1∆Cyt were challenged with CHIKV strains of ECSA 3′GFP-CHIKV (MOI = 0.01), WA 5′GFP-CHIKV (MOI = 0.01) or Asian mc-CHIKV genotypes (MOI = 0.1) and infection assessed by flow cytometry at 24 hpi. (**D**) Cold binding of fluorescent mc-CHIKV at MOI of 50. After two hours and extensive washes, cells were fixed and analyzed by flow cytometry. (**E**) Live cell imaging of CHIKV binding. Cells were inoculated with fluorescent Asian mc-CHIKV at MOI of 50 and monitored by confocal microscopy. Number of CHIKV binding events to the plasma membrane within 20 min were counted. The error bars represent standard errors of the mean (SEM). Statistical significance was calculated using Dunnet’s multiple comparisons test (2way ANOVA) (**C**) and Welch *t*-test (**E**) with ns > 0.05, * *p* < 0.05 and **** *p* < 0.0001.

**Figure 3 cells-10-01828-f003:**
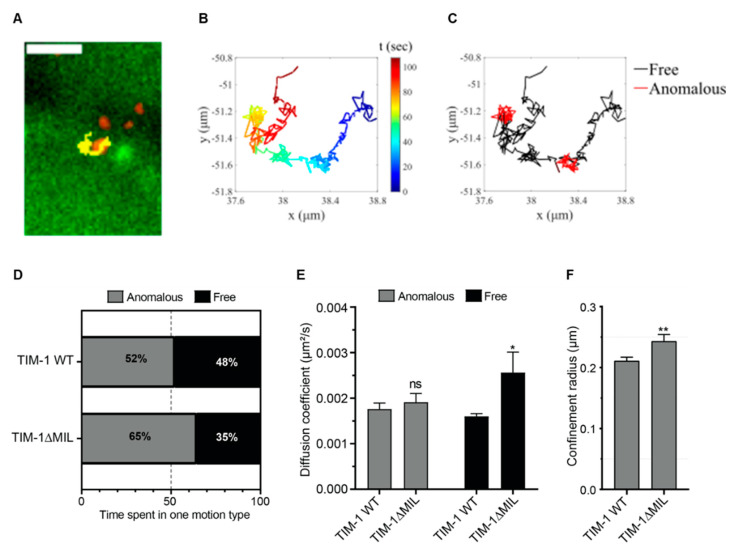
Single particle tracking of CHIKV confirms PtdSer domain requirement. (**A**) Overlay of the fluorescence image of CalceinAM stained TIM-1 WT cells (green) and labeled Asian mc-CHIKV particles (red) with the virus diffusion trajectory (yellow) at the cell surface. The bar represents 2 μm. (**B**) The CHIKV trajectory shown in (**A**) as a time-lapse with the time in seconds presented as a color bar (**C**) Segmentation of the trajectory shown in (**B**) and classification of the segments using moment scaling spectrum (red: anomalous confined motion—black: Brownian free motion). (**D**) The fraction of time spent by the mobile particles either in confined anomalous or Brownian free motion. The total number of viruses analyzed here is N_virus_ = 1523, 472 for TIM-1 WT and TIM-1 ΔMIL respectively. (**E**) The mean of the diffusion coefficient for anomalous and free motion of each segment is calculated using MSS as described in methods. (**F**) The confinement radius of the anomalous motion is presented for each cell type. The error bars represent standard errors of the mean (SEM) with N_segments_ = 1296, 414 for TIM-1 WT and TIM-1ΔMIL respectively. Statistical significance was calculated using a Welch *t*-test with ns > 0.05, * *p* < 0.05 and ** *p* < 0.01. For D, the percentage of fraction was obtained by counting, so no statistical analysis is provided.

**Figure 4 cells-10-01828-f004:**
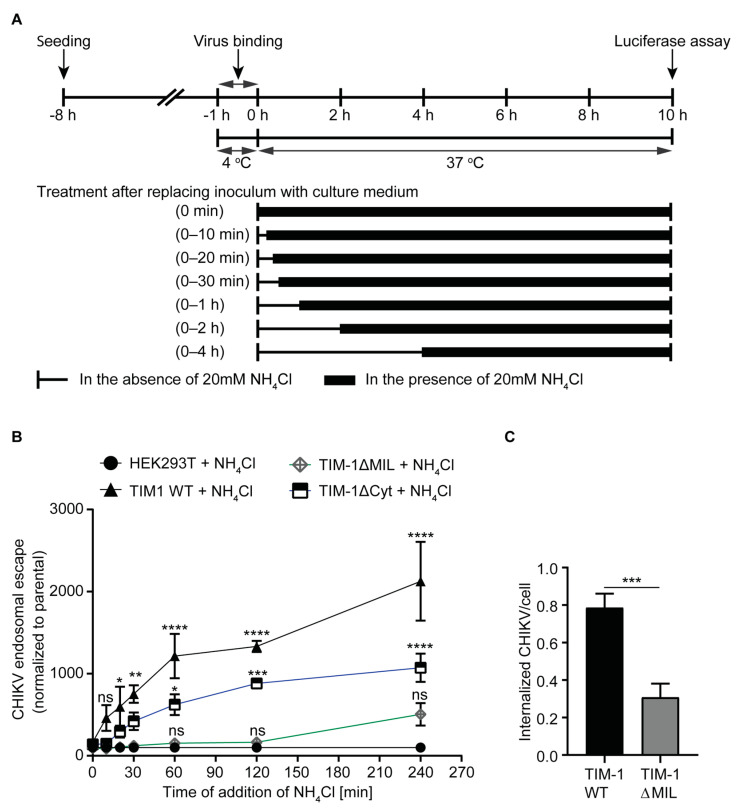
Entry kinetics of CHIKV are altered by TIM-1. (**A**) Scheme showing binding and entry assay. CHIKV (Asian genotype) encoding nano-luciferase fused to E2 glycoprotein (nLuc-CHIKV) was added to parental and TIM-1 expressing HEK293T cells, which were subsequently incubated at 4 °C for 1 h to synchronize binding. After washing, cells were transferred to 37 °C and medium with 20 mM NH_4_Cl was added at indicated time points. Assay was stopped after 10 h of incubation at 37 °C and relative endosomal escape determined by luciferase assay. (**B**) Entry kinetics of nLuc-CHIKV in the indicated cell lines normalized to parental HEK293T cells at each time point. (**C**) Live cell imaging of internalized CHIKV. Cells were inoculated with fluorescent Asian mc-CHIKV at MOI of 50 and monitored by confocal microscopy. After 20 min of live imaging of fluorescent mc-CHIKV (Asian genotype) and cells at 37 °C, trypan blue was added to quench extracellular particles and only internalized viruses were imaged and counted using ImageJ. The error bars represent standard errors of the mean (SEM). Statistical significance was calculated using Dunnet’s multiple comparisons test (2way ANOVA) (B) and Welch *t*-test (C) with * *p* < 0.05, ** *p* < 0.01, *** *p* < 0.001 and **** *p* < 0.0001.

**Figure 5 cells-10-01828-f005:**
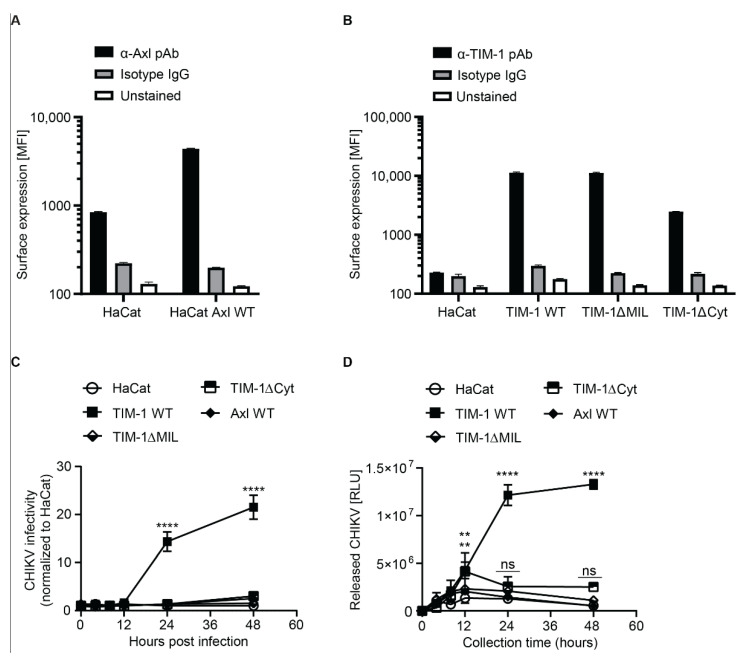
TIM-1 renders HaCat cells permissive to CHIKV. (**A**) Cell surface expression of Axl in HaCat cells with and without ectopic Axl expression (Axl WT) analyzed by antibody staining and flow cytometry. (**B**) Cell surface expression of ectopic TIM-1 WT and TIM-1 mutants in HaCat cells analyzed by antibody staining and flow cytometry. (**C**) Parental HaCat immortalized keratinocytes and HaCat cells expressing either TIM-1 WT, TIM-1ΔMIL, TIM-1ΔCyt or Axl WT were inoculated with Asian genotype nano-luciferase CHIKV reporter virus. After four hours, the cells were washed extensively to remove unbound virus. Expression of nano-luciferase attesting for viral replication was monitored over time using luciferase assay. Permissive fold change relative to parental HaCat cells at the indicated time points post infection is shown. (**D**) Released progeny virions in culture supernatants from (**C**) were used to inoculate human dermal fibroblasts and infection levels determined at 24 h post infection. The error bars represent mean ± SEM of three independent experiments. Statistical significance was calculated using a Dunnet’s multiple comparisons test (2way ANOVA) ns > 0.05, ** *p* < 0.01 and **** *p* < 0.0001.

**Figure 6 cells-10-01828-f006:**
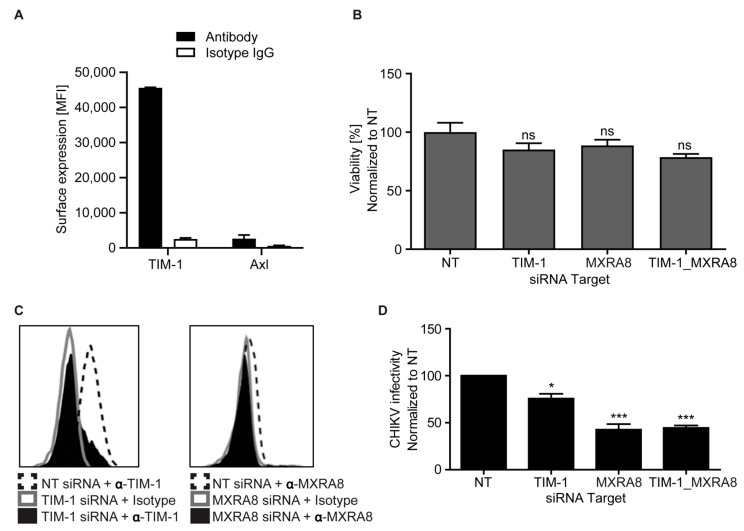
Endogenous expression of TIM-1 facilitates CHIKV infection of hepatocytes. (**A**) Endogenous surface expression of TIM-1 and Axl in Huh7.5 cells. (**B**) MTT assay-based viability of Huh7.5 cells. Cells were re-seeded in 96-well plate 48 h after treatment with targeting and non-targeting (NT) siRNA and incubated overnight. Afterwards, MTT assay was performed to compare proliferation of cells. (**C**) TIM-1 and MXRA8 surface expression 48 h after siRNA treatment measured by antibody staining followed by flow cytometry. One representative dataset shown. (**D**) Huh7.5 cell susceptibility to CHIKV after treatment with targeting and NT siRNA. Cells were inoculated with 5′GFP CHIKV (WA genotype) and infectivity determined by flow cytometry 24 hpi. The error bars represent mean ± SEM of at least three independent experiments. Statistical significance was calculated using a Dunnet’s multiple comparisons test (2way ANOVA) ns > 0.05, * *p* < 0.05 and *** *p* < 0.001.

## Data Availability

Not applicable.

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
