# Peer review of "The Phosphatidylserine Receptor TIM-1 Enhances Authentic Chikungunya Virus Cell Entry"

_cells, 2021, doi:10.3390/cells10071828_

Round 1
Reviewer 1 Report
This manuscript examines Chikungunya virus’ use of phosphatidylserine (PS) receptors for entry into cells. While others have previously shown that entry of pseudovirions bearing CHIKV E is enhanced by PS receptors, detailed studies with infectious virus has not previously been performed. The authors focus on two of what are thought to be the more important PS receptors: TIM-1 and AXL. The authors report that TIM-1, but not AXL, enhances virus infection in these studies which are primarily (although not exclusively) overexpression studies. The studies support previous studies indicating that the TIM-1 PS binding pocket is important for TIM-1 dependent uptake. The authors perform single particle tracking studies that appear to provide somewhat conflicting data. In addition other reported findings also have internal inconsistencies and the authors simply present those inconsistencies with little or no interpretation/elaboration of the findings. Conclusions drawn from those studies do not necessarily take into account the inconsistencies reported.
Experimental concerns:
S1: why are doublets present in western blots of AXL and TIM-1? The size differences in the anti-TIM-1 bands suggests that the higher band may be a dimer. Is there evidence that TIM-1 exists in cells as a dimer? What are the two AXL bands that are separated by ~30 kDa?
Fig. 1E: Is the modest inhibition observed with anti-TIM-1 statistically significant?
S2: The data shown suggest that in the absence of GAGs, TIM-1 dependent entry is reduced. Is the reduction statistically significant? How different are the baseline values (ie, infection in the two lines in the absence of TIM-1)? As the data are currently shown, it is not possible for the reader to determine this since all baseline values are set to 1. The conclusion the authors draw is that TIM-1 dependent infection is “largely independent” of GAGs. This conclusion does not appear to be consistent with the findings as there appears to be a component of TIM-1 dependent entry that is impacted by the loss of GAGs.
Fig. 2: Why use the Asian strain of virus to assess binding of TIM-1+ cells since this strain had lower levels of infectivity? The statistically indistinguishable CHIKV binding shown in Fig. 2D suggest that other residues within TIM-1 contribute to CHIKV binding, but this is ignored/overlooked (?) by the authors. Findings in Fig. 2D and E are not internally consistent (D shows there is no difference in CHIKV binding, whereas E shows that number of particles bound are different between cells expressing WT TIM-1 and a line expression a TIM-1 PS pocket mutant). Please discuss the different in these findings. Also details of the internalization assays are needed in the M&M section.
Fig. 3: The authors show in single particle imaging studies that CHIKV spent less time diffusing freely on cells expressing the PS binding pocket mutant than WT TIM-1 (Fig. 3D) yet with a higher diffusion coefficient. The authors overall conclude from these studies that the PS binding site of TIM-1 contributes to slow down the lateral virus movement at the plasma membrane, yet reduced mobility of the mutant compared to WT TIM-1 is not consistent with this interpretation. Perhaps additional controls that include comparing movement of other viruses that have high affinity surface receptors would prove useful.
Fig. 4: From the data presented, the authors conclude that the cytoplasmic tail of TIM-1 is needed for CHIKV internalization, but the data do not support this strongly stated conclusion. The rate of endosomal escape by the cytoplasmic tail mutant may be delayed, but biological variation found in these experiments is not shown. Further, the entry via TIM-1 or mutants is currently compared to that of virus in parental cells. As these cells are poorly infectable, this seems like an odd comparator. The rationale for using this control as a comparator is not presented. Is the difference in time of virus entry in these different cells statistically significant? Further since the earlier figures indicate that virus infection supported by WT and the cytoplasmic tail mutant are indistinguishable, the effect of the loss of the tail on overall virus infectivity must be modest at best.
Fig. 6: Shown are overexpression studies in HaCat cells. The rationale behind these studies is exploration of AXL and TIM-1 dependent entry into a relevant cell type, keratinocytes. However, the use of TIM and AXL overexpression studies are used here and call into question the rationale for the use of this line. Within the discussion, the authors state that data from these studies support the idea that TIM-1 is a key factor for CHIKV infection in the skin (Line785). I would argue that this is an over interpretation of the data. These data indicate that TIM-1 can be overexpressed when a TIM-1 construct is delivered into this cell line and exogenous TIM-1 expression is functional in these cells.
In these cells, endogenous AXL is expressed. The authors introduce WT or mutant AXLs into the cells without knock out of endogenous AXL. This results in as much as 5-fold higher AXL or mutant AXL expression. The authors find that added expression of AXL does not enhance pseudovirion entry. As ascribed by the authors, this may be due to the inability of the virions to utilize AXL as a receptor, but what is not excluded here is an alternative possibility that AXL expression is already providing some entry into the cells and that addition of more AXL is not enhancing entry. Without knocking out the endogenous AXL expression, it is difficult to parse out what is occurring. Also it is difficult for the reader to determine what the baseline of infection is. While I appreciate that normalizing virus infections to a baseline allows pooling of datasets and ready comparison, it also does not provide evidence of baseline permissivity of the cell. Perhaps a table providing baseline levels of infection for the different cell lines used would be one simple way to alleviate this problem.
Fig. 7: The authors use human dermal fibroblasts as a titering line. Entry of other alpha and flaviviruses into human fibroblasts is dependent upon AXL (eg, doi: 10.1128/JVI.00354-15). Thus, it is likely that the authors are evaluating the virus produced in HaCat cells by titering via a line dependent upon AXL entry – which may be skewing the findings given that that this manuscript shows that CHIKV poorly utilizes AXL. Studies to determine whether AXL is being used in these titering studies is needed.
Fig. 8: Given that knock down of MXRA8 and the combination of knock down of MXRA8 and TIM-1 yield similar virus decrease, it would be of value to determine if TIM-1 expressed on a subset of MXRA8 expressing cells. Also it appears that low levels of AXL are expressed on these cells (staining is above the isotype control level). Do AXL inhibitors block infection?
Other comments:
On line 843, the authors suggest the possibility that authentic CHIKV may be utilizing TIM-1 in a manner other than through PS/PS receptor interactions. Given the TIM-1 ΔMIL mutant abrogates enhanced infection, the current studies support that PS/TIM-1 interactions are important for CHIKV infection and do not support this speculative contention.
Reviewer 2 Report
In the manuscript Kirui et al. the authors analyze the role of phosphatidylserine-dependent receptors, namely TIM-1 and AxI for infection of CHIKV in different cells (HEK293T, HaCat and Huh7.5 cells). The experiments were performed using cells stably expressing the respective surface proteins using lentiviral vectors (HEK293T, HaCat) or knock-down experiments (Huh7.5 cells). After demonstrating a role for TIM-1 in CHIKV infection, the authors established TIM-1 variants (mutation of PtdSer-binding pocket and deletion of cytoplasmic domain) to dissect the role of these TIM-1 domains for CHIKV infection. Infection experiments were performed using reporter viruses (GFP, mCherry, nLuc) of the different CHIKV genotypes. After analyzing the effect of overexpression of TIM1-1 and variants on CHIKV infection, studies to analyze the effect on RNA replication, virus binding and internalization were included as well.
Overall, the manuscript contains interesting data and methods. However, the manuscript would benefit from some reorganization like changing the order of described experiments or moving certain data into the supplements, since the paper is also rather lengthy and sometimes little difficult to follow.
Regarding reorganization – the data with the replicon seem quite lost where they are (lines 464 ff). What about moving it after the first infection experiments. Than argue that since there is no difference in replication levels, the difference might be on the level of entry and/ or internalization and continue with these experiments.
Also regarding reorganization and more important: it is quite confusing that there is a section on binding and internalization in section 3.2 (Lines 433 ff / Fig. 2D, E,F) and again in section 3.4 (Fig. 4B, C).
Further major comments:
- Abstract: the HaCat results are not mentioned at all -this should be done especially since cell type specific results occur.
- Line 32: Unlike Ax1 – but there was a 10 fold difference described (line 352)?
- Line 111 and results section on binding: please mention where the mCherry is inserted in the mCherry virus and add the respective reference. This is important since the binding assay with this virus only makes sense when it is in the envelope region and ends up in the particle. Similar – describe the CHIKV nLuc virus better (seems to also include luciferase).
- The authors use the terms entry, internalization, endosome escape. Should entry be the same as internalization here? 2.9 only describes binding but nothing for entry/ internalization. Only endosomal escape, which would be after internalization. Please add method for internalization.
- Line 457: After 20 min of what? Was there first incubation at 4°C and then shift to 37°C – shift is not mentioned. For what is trypan blue used here?
- In line with point 1 – it would be helpful if at least in the figure legends the viruses are indicated including their reporter: e.g.line 3789: ECSA CHIKV-3’-GFP,… Asian = mCherry? Or for VEE= VEE-GFP in legend S2B.
- Move Fig. 6 to supplements (the story is more about ‘authentic’ virus – and Fig. 6 are pseudotype viruses).
- Discussion: line 688 ff: again: is 10 fold increase for Ax1 really negligible? Please mention the data (how much increase) that was published in the previous studies mentioned (using pseudotyped viruses).
Minor points:
- Line 155: concentration for blasicidin selection
- Line 184: Provider Cap Analog, SP6 Polymerase
- Line 191: in vitro
- Line 192: Opti-MEM nocht OptiMem
- Line 203 ff: cell numbers seeded for cell lines
- Line 260: briefly mention what Calcein AM staining is good for
- section 2.14: were cells fixed before antibody staining?
- TIM-1, Ax1 and CHIKV: abbreviations have already been introduced before
- line 360: Moi 0.01 not O.01
- Line 378 Asian = mCherry virus
- 11: line 400: indicate fold change instead of 26% (for better comparison to other values)
- Line 421: Add Figure referene Fig. 2C for TIM1-deltaMIL data.
- Legend Fig. 2A: Representation of cells stably expression: there are no cells shown – rather schematic drawing of wt and mutant TIM…
- Leg. Figure 5: A): Endogenous? B) Ectopic?
- line 594: keratinocytes: are this the HaCat cells? Than please change accordingly.
- line 624: Fig. reference Fig. 7
- line 628ff: inoculation for 24 h – and then what? Measure reporter to determine viral growth?
- Labeling x axis Fig. 7b: Simply RLU or maybe released CHIKV (RLU) (- infectivity does not really make sense).
- Text and legend/labeling in Figure8: sometimes it is Mxra8, sometimes MXRA8
- Legend Fig. S1: Transduction of cells with lentiviral particles
Round 2
Reviewer 1 Report
The manuscript by Kirui et al is improved. With changes in two areas, this is now ready for publication.
- In the first Table, there are two rows labeled CHO K1-TIM-1, but there is no row labeled CHO 745 TIM-1. Is one of the rows mislabeled?
- Experimental details associated with Figure 4 remain sketchy in both the Results and M&M sections. I believe that the virus particles express both nano-luciferase attached to the glycoprotein and luciferase that is being expressed from the viral genome, but that is not entirely clear. Please clarify.
